# Married women decision making autonomy on health care utilization in high fertility sub-Saharan African countries: A multilevel analysis of recent Demographic and Health Survey

**Wubshet Debebe Negash**[1]*, **Getachew Teshale Kefale**[1], **Tadele Biresaw Belachew**[1], **Desale Bihonegn Asmamaw**[2]

**1** Department of Health Systems and Policy, Institute of Public Health, College of Medicine and Health Sciences, University of Gondar, Gondar, Ethiopia, **2** Department of Reproductive Health, Institute of Public Health, College of Medicine and Health Sciences, University of Gondar, Gondar, Ethiopia

* wubshetdn@gmail.com

**Data Availability Statement:** Data for this study were sourced from Demographic and Health

## Abstract

### Background

Women's decision-making autonomy has a potential impact on the scale-up of health care utilization. In high fertility countries, evidence regarding women's decision-making autonomy on their health care utilization and its associated factors is limited and inconclusive. Hence, it is important to investigate women decision-making autonomy on their health care utilization and associated factors in high fertility countries in sub-Saharan Africa.

### Methods

The data source for this study was obtained from recent Demographic and Health Surveys that were comprised of a weighted sample of 178875 reproductive age women. A multilevel mixed-effect binary logistic regression model was fitted. The odds ratios, along with the 95% confidence interval were generated to identify individual and community-level factors associated with women's autonomy in health care decision-making. A p-value less than 0.05 was declared as statistical significance.

### Results

In this study, 42% (95% CI: 41.7, 42.3) of women were able to exercise their reproductive autonomy. The highest (74.8%) and the lowest (19.74%) magnitude of women autonomy was found in Angola and Mali, respectively. In multilevel analysis; age of women 25–34 years, 35 and above (AOR = 1.34, 95% CI: 1.29, 1.39), and (AOR = 1.78, 95% CI: 1.75, 1.90), women's primary and secondary educational level (AOR = 1.25, 95% CI: 1.20, 1.31), and (AOR = 1.44, 95% CI: 1.32, 1.54), husband primary and secondary educational level (AOR = 1.24, 95% CI: 1.18, 1.29), and (AOR = 1.21, 95% CI: 1.15, 1.27), women who had

surveys (DHS), which is freely available online at (https://dhsprogram.com). To request data for addressing similar or another research questions, the request should be submitted to the DHS program via the link: https://dhsprogram.com/data/Access-Instructions.cfm. The DHS Program will typically evaluate all the requests within one to two days (during working days) and notify if access has been allowed or if additional project evidence is required before access can be granted. Following permission, the researcher able to login and choose the specific data in the format of their choice.

**Funding:** The authors received no specific funding for this work.

**Competing interests:** The authors declare that they have no competing interests.

**Abbreviations:** AOR, Adjusted Odds Ratio; CI, Confidence interval; DHS, Demographic and Health Survey; ICC, Intra-class Correlation Coefficient; MOR, Median Odds Ratio; PCV, Proportional Change in Variance; SD, Standard Deviation; SSA, Sub-Saharan Africa; WHO, World Health Organization.

work (AOR = 1.67, 95% CI: 1.59, 1.74) female household heads (AOR = 1.44, 95% CI: 1.37, 1.51), media exposure (AOR = 1.04, 95% CI: 1.09, 1.18), health insurance coverage (AOR = 1.26, 95% CI: 1.17, 1.36), urban residence (AOR = 1.14, 95% CI: 1.09, 1.19), community education (AOR = 2.43, 95% CI: 2.07, 2.85) and low community poverty level (AOR = 1.27, 95% CI: 1.08, 1.49) were predictor variables.

## Conclusion and recommendation

Although every woman has the right to make her own health care decisions, this study showed that almost 58% of them had no role in making decisions about their health care utilization. Thus, each country Government should support women's decision making autonomy regarding their healthcare utilization through mass media and extensive behavioral education.

## Background

Autonomy is the ability to gather knowledge and make judgments regarding personal matters [1]. The feminists activists defined women's autonomy as a belief in and advocacy of equal rights for women based on the idea of the equality of the sexes [2]. Relational definition, on the other hand, incorporates insights into socially embedded notions such as interactions with family, community, and people [3].

Women's autonomy is usually measured by four dimensions: gender division of labor, norms, control over assets, and decision-making [1]. The ability of women to seek health care services and visit health facilities depend on their autonomy of decision-making. A woman's autonomy in making health care decisions is mandatory for protecting maternal and child health outcomes [4]. It is integral to each of the 17 goals of Agenda 2030 for sustainable development [5].

Findings implied that in Ethiopia, Nigeria and Ghana, reducing women's independence through restricting their decision-making and financial ownership results in reduced utilization of healthcare services [4, 6, 7]. Despite better independence for women in maternal health care consumption, women in low-income countries have minimal regulator over home assets and involvement in health care decisions [8, 9]. In developed nations, the use of maternal health care is primarily driven by women's autonomy [1], which delivers multiple benefits such as lower fertility rates and more resources given to children in the home [10]. Women are usually considered to have less governor over assets and involvement in healthcare decisions in African countries [11].

Women's decision with regard to the use of healthcare services may depend on their socio-economic position, perception towards modern healthcare treatment, family size, perceived severity of illness, and previous experience of illness [12, 13]. The aforementioned factors have frequently demarcated the situations under which women did or did not have autonomy to involve in decisions regarding their health. Additionally, both individual-level and community-level autonomy can be a source to influence a woman's decision to seek healthcare [14]. At the individual level, low autonomy can affect women's health through fewer opportunities to engage in paid employment, the presence of domestic violence, and restricted contact to health care services [15, 16].

Women are prohibited from leaving their homes for health care in the most countries of SSA [17]. A study in rural SSA countries showed that women living in communities are not

expected to visit health care facilities, which reduces the Service utilization in the health care system [14]. In developing countries, women's autonomy is the most important factor influencing maternal health care utilization [1]. The use of maternal health for care is influenced by the autonomy of women within the household in making decisions [4, 18].

Although multiple researches were tried to assess the magnitude of women's decision-making autonomy on health care utilization, most of them concerned a single country or a limited geographical area and specific health care services [4, 13, 18]. There is limited evidence of the pooled magnitude and associated factors of healthcare decision-making autonomy among women in high fertility SSA countries. The determination of the magnitude and identification of the factors affecting women's autonomy in health care decision making will be essential for increasing maternal healthcare utilization, and to minimize the morbidity and/or mortality. Therefore, this study is very important because the result will be useful as an input for program planners and resource allocators to advance women health. It also showed the countries' standing in relation to the Millennium Development Goals (MDGs) and Sustainable Development Goals (SDGs) of women's empowerment and encouraged them to take action for their achievement. This study was conducted to assess the magnitude of women's decision-making autonomy on healthcare use and its factors among married women in selected high-fertility SSA countries.

## Methods

### Study design

A community-based cross-sectional survey with a quantitative method was conducted between January 2010 and December 2018 among married reproductive-age women.

### Study settings

The survey included nine SSA countries: Niger, Democratic of Republic Congo, Mali, Chad, Angola, Burundi, Nigeria, Gambia, and Burkina Faso. These countries were selected due their fertility rates above 5.0, making them high fertility countries in SSA. This fertility rate is higher than the rates in sub-Sahara Africa (4.4) and worldwide (2.47) [19]. Somalia, one of the SSA nations with a high fertility rate, was left out of the research since there were no DHS data.

### Data sources

The data for these countries were obtained from the official database of the DHS program, https://dhsprogram.com after authorization was granted via an online request explaining the purpose of our study. Our analysis was based on the women's records (IR files) data set, and we extracted the dependent and independent variables. Every five years, DHS takes a nationally representative household survey throughout low- and middle-income countries. It has been an essential data source on issues of reproductive health in low and middle income countries as it gathers data on a number of reproductive health issues [20].

A two-stage stratified-cluster sampling method was used by the DHS. Within each sampling stratum, proportional allocation was accomplished prior to sample selection at various levels. In the initial stage, each sampling stratum was chosen from the available samples, and enumeration areas (EAs) were chosen with a probability proportional to their size. The number of residential units in each EA was determined by household listing procedures. The lists of households that were produced after that were then utilized to pick the households for the sampling frame. Households from each cluster were randomly chosen in the second stage. Only the pre-selected families were subjected to interviews. On the website www.measuredhs.

com, you can get more details regarding the sample plan or technique. 178875 women (15–49 years old) of reproductive age were included in the study's weighted total sample (Table 1).

## Variables and measurements

**Outcome variable.** The autonomy of women in making decisions about how to use healthcare services was the outcome variable. Based on their vocal comments during the survey, the DHS gathered information on women's decision-making autonomy in using healthcare services. The information gathered focused on whether the woman makes decisions about her use of healthcare alone, jointly with their partner, jointly with another person, or with someone else. Based on existed literatures [21, 22] and for the analysis purpose, the outcome was dichotomized as "not autonomous" = 0 (for married reproductive age women who reported that the decision regarding their use of healthcare was made primarily by their partner alone or by someone else) and "autonomous" = 1 (for married reproductive age women who reported that the decision regarding their use of healthcare was primarily made by the respondent alone and/or jointly with their partners).

## Independent variables

Independent variables were considered in this study to determine factors associated with women's decision-making autonomy in health care utilization. These independent variables were chosen based on experts' experience, their importance on the outcome variable, and a review of similar published articles [18, 22–24]. These variables were grouped as characteristics of women, family/household, and community-level factors. The woman's characteristics include age, educational level, and occupation. The family/household characteristics were husband education, media exposure, wealth index, health insurance, household size, and sex of the household head, whereas residence, community level media exposure, community level poverty, and community level education were considered as key community level variables.

Accordingly, the age of the women was grouped as 15–24, 25–34, and 35–49 years, whereas no formal education, primary education, secondary education, and higher education were categories for the woman's and her husband's education. Occupational status was grouped as working and not working, while having health insurance was categorized in the DHS report as yes or no. Household family size was categorized as 1–2, 3–5, 6 and above. A "household head" is a person acknowledged by the household members as taking the main role in the health care decision-making process of a family and responsibility for its survival [25]. In this study, the sex of the head of the household was categorized as female or male. The media exposure status was based on whether a woman watches television, listens to the radio, and reads a newspaper

**Table 1. Description of surveys and sample size characteristics in high fertility countries in SSA (n = 178875).**

| Countries 3.19 | Survey year | Weighted sample(n) | Weighted sample (%) |
|---|---|---|---|
| Angola | 2015/16 | 7957 | 7.17 |
| Burkina Faso | 2010 | 13563 | 12.22 |
| Burundi | 2016/17 | 9782 | 8.81 |
| DR Congo | 2014/15 | 12096 | 10.90 |
| Chad | 2013/14 | 13262 | 11.95 |
| Gambia | 2013 | 6791 | 6.12 |
| Mali | 2018 | 8567 | 7.72 |
| Nigeria | 2018 | 29089 | 26.21 |
| Niger | 2012 | 9881 | 8.90 |

or magazine. A woman who has one or more exposures has been considered to have media exposure. In the DHS, the household wealth index was grouped as poorest, poorer, poor, richer, and richest. As a result of the high variability of observation in the original DHS classification of households into five categories using principal component analysis, the wealth index scores were re-categorized into three categories (poor, medium, and rich) by merging the poorest with the poorest and the richest with the richest for the ease of interpretation of principal component analysis. A detailed description of all variables was provided in the 2016 DHS report [26].

Community-level variables such as type of residence, country, and distance to the health facility were directly accessed from the DHS data. On the other hand, community-level factors of community level media exposure, education, and poverty were created by aggregating individual-level characteristics of the study clusters. These community level variables were then, categorized as high and low based on the distribution of the proportional values computed for each variable. The aggregate variables were not normally distributed, and the median value was used as a cut-off point for the categorization. Accordingly, community level media exposure, community level education, and community level poverty were categorized as high if the proportion was greater than 50% and low if the proportion was less than or equal to 50% based on the national median value since these were not normally distributed [27].

## Data analysis

Stata version 14 was used for data analysis. To ensure the representativeness of the DHS sample and provide accurate estimates and standard errors, the data were weighted during the study. The data from the demographic and health surveys are arranged hierarchically, with mothers nested within a group. The independence and equal variance assumptions of conventional logistic regression may not apply in this situation. A multilevel binary logistic regression model was constructed to account for the hierarchy. In a cluster of women, there may be similarities in their characteristics. Since observations are not independent and the variance is not equal between clusters, an advanced model must be employed to account for the differences between clusters. Therefore, a fixed effect for factors and a random effect for the between cluster variation were estimated using a two-level mixed-effect logistic regression analysis, by assuming that each community has a different intercept and fixed coefficient, with a random effect applied at the cluster level [28, 29].

Accordingly, the multilevel logistic regression model was fitted using the equation:

$$\text{Log}\left[\pi ij/(1 - \pi ij)\right] = \beta 0 + \beta 1 xij + \beta 2 xij \ldots + \mu 0j + e0ij$$

Where: $\pi ij$: The probability of women autonomy for health care decision

$1 - \pi ij$: The probability of no autonomy for health care decision

$\beta_1 xij$: individual and community level factors for the $i^{th}$ individual in group j, respectively.

$\beta$'s: are fixed coefficients representing a unit increase in X can cause a β unit increase in probability women autonomy in health care decision.

$\beta_{0:}$ is intercept that is the effect on women autonomy in health care decision when the effect of all factors are absent.

Uj: indicates the random effect for the $j^{th}$ community [30, 31].

In this research, four models were fitted: the null model, which didn't include any independent variables; model I, which included factors at the individual level; model II, which contained factors at the community level; and model III, which comprised elements at both the person and community levels. The intra class correlation coefficient (ICC) and median odds ratio (MOR) were calculated as follows: ICC [32] = $ICC = \frac{V_A}{V_A + \frac{\pi^2}{3}}$ Where VA means the community

level variance $\frac{\pi^2}{3}$ represents the individual level variance [32]. On the other hand MOR was determined by the formula: exp.$[\sqrt{(2 \times VA)} \times 0.6745]$. For this study VA reveals the cluster level variance. The value 0.6745 is the 75th percentile of the cumulative distribution of the normal distribution function [33]. The intra-class correlation coefficient (ICC) and median odds ratios (MOR) were used to compare models, while deviances (-2LLR) were used to evaluate the models' fitness. The model with the lowest deviation, Model III, was selected as the best fit.

To explain the variation in health care decision-making autonomy between study subjects, proportional change in variance (PCV) was calculated by the formula: PCV; [(Vnull-$V_A$)/Vnull]*100, where; Vnull is community variance of model without covariates (null model) and $V_A$ is community variance in the models including individual, community or both individual and community-level factors [33].

For multivariable analysis, variables having a p-value of less than 0.2 in bi-variable were utilized to reduce confounding. In the multivariable study, characteristics related to women's autonomy in decision-making were identified using adjusted odds ratios with 95 percent confidence intervals and a p-value less than 0.05.

### Ethical approval and consent to participate

The ethical approval and permission to access the data were freely obtained from the DHS website www.measuredhs.com. During the original collection of DHS data, international and national ethical guidelines were taken into account. Ethical clearance for the original DHS was approved by the ICF Macro Institutional Review Board, the Centers for Disease Control and Prevention (CDC) and Institutional Review Board (IRB) in each country, in accordance with United states Department of Health and Human Services requirements for human subject protection. According to the DHS, all respondents and/or their legal guardian(s) of minors' (age below 16) provided written consent to participate. All the methods were conducted according to Helsinki declarations. No information obtained from the data set was disclosed to any third person. The study is not an experimental study. Further explanation of how the DHS uses data and its ethical standards can be found at: http://goo.gl/ny8T6X.

## Results

### Socio-demographic, obstetric and health service accessibility related characteristics of the study participants

Overall weighted sample of 178875 reproductive age women participated to this study. The majority of study participants (40.49%) were classified as having low wealth status, with 69.32% living in rural areas. Over half of the respondents were not educated (55.05%) and had media exposure (61%). The majority (88.48%) of participants' household heads were males. Almost all (94.98%) women's health care services were not covered by health insurance (Table 2).

### Women's autonomy in decision making

The overall magnitude of women's decision-making autonomy on health care utilization in high-fertility SSA countries was 42.03% (95% CI: 41.74, 42.32). The lowest (19.74%) and the highest (74.8%) decision-making autonomy were in Mali and Angola, respectively (Fig 1).

### Multi-level analysis of factors

Based on the final model result, the age and educational level of the women, the husband's educational level, the women's occupational status, the sex of the household head, media

**Table 2.  Socio-demographic, obstetric and health services accessibility characteristics of women in high fertility sub-Saharan Africa countries (n = 178875).**

| Variables | Category | Frequency | Percent |
|---|---|---|---|
| Age in years | 15–24 | 28125 | 25.34 |
| | 25–34 | 44566 | 40.15 |
| | 35+ | 38297 | 34.51 |
| Residence | Urban | 34047 | 30.68 |
| | Rural | 76941 | 69.32 |
| Educational status of respondents | No education | 61099 | 55.05 |
| | Primary education | 24061 | 21.68 |
| | Secondary & Higher education | 25828 | 23.27 |
| Husband education | | | |
| | No formal | 51897 | 46.76 |
| | Primary | 19852 | 17.89 |
| | Secondary and higher | 39239 | 35.35 |
| Occupation of respondents | Not working | 27290 | 24.59 |
| | Working | 83698 | 75.41 |
| Wealth | Poor | 44935 | 40.49 |
| | Middle | 2471 | 20.25 |
| | Rich | 43582 | 39.27 |
| Mass media exposure | Yes | 67694 | 61.00 |
| | No | 43294 | 39.00 |
| Distance to the health facility | Big problem | 38314 | 34.52 |
| | Not big problem | 72674 | 68.48 |
| Sex of household head | Male | 98204 | 88.48 |
| | Female | 12784 | 11.52 |
| Covered by health insurance | Yes | 5132 | 5.02 |
| | No | 97148 | 94.98 |
| Household size | 1–2 | 5768 | 5.20 |
| | 3–5 | 40681 | 36.65 |
| | 6 and above | 64539 | 58.15 |

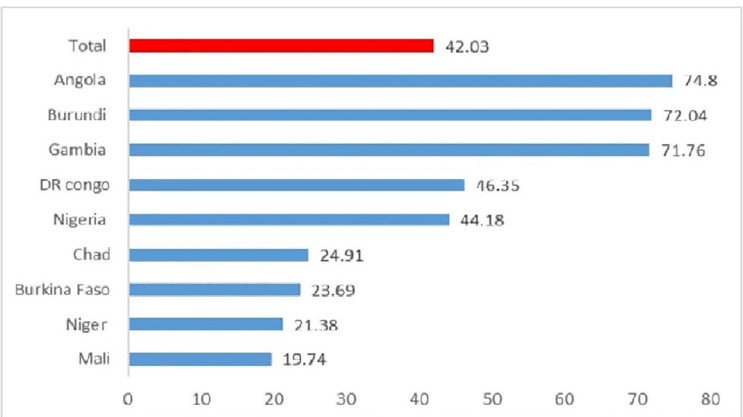

**Fig 1. Women's decision making autonomy on health care utilization in high fertility sub Saharan Africa countries.**

exposure, health expenditure covered by health insurance, residence, and community education were significantly associated with women's decision-making autonomy on their health care utilization.

When compared to younger women (15–24 years), women aged 25–34 years and 35 years and older had 1.34 (AOR = 1.34 95% CI: 1.29–1.39) and 1.82 (AOR = 1.82, 95% CI: 1.75–1.90) times higher odds of decision-making autonomy on health care utilization, respectively.

Women who had primary education and secondary education had 1.25 (AOR = 1.25, 95% CI: 1.20, 1.31), and 1.44 (AOR = 1.44, 95% CI: 1.32, 1.54) times higher odds of decision-making autonomy on their health care utilization as compared to those women who did not have education, respectively.

The odds of women's decision-making autonomy on health care utilization among women whose husbands had primary and secondary education levels were 1.24 (AOR = 1.24, 95% CI: 1.18, 1.29) and 1.21 (AOR = 1.21, 95% CI: 1.15, 1.27) times higher than those whose husbands did not have education, respectively.

The odds of participating in making decisions about their health care utilization were 1.67 times higher among women who had work (AOR = 1.67, 95% CI: 1.59, 1.74) than their counterparts.

The odds of women participating in decision-making for their health care utilization were 1.44 times higher among female household heads (AOR = 1.44, 95% CI: 1.37, 1.51) as compared to male household heads. Women who had media exposure had 1.04 (95% CI: 1.09, 1.18) times higher odds of participating in decision-making for their health care utilization than their counterparts. The likelihood of women participating in their health care utilizations was 1.16 (AOR = 1.26, 95% CI: 1.17, 1.36) times higher among those with health insurance than among those without.

Women who lived in urban areas were 1.14 (AOR = 1.14, 95% CI: 1.09, 1.19) times more odds of participating in decision making for their health care utilization as compared to women who lived in rural areas. The odds of participating in decision making for their health care utilization was 2.43 times (AOR = 2.43, 95% CI: 2.07, 2.85) higher among in a community with high education than their counter groups and 1.27 (AOR = 1.27, 95% CI: 1.08, 1.49) times higher among in a community with low poverty level than community with high poverty level. Women in Angola, Burundi, DR Congo, Gambia, and Nigeria were 6.44 (AOR = 6.44, 95% CI: 5.87, 7.06), 5.69 (AOR = 5.69, 95% CI: 5.21, 6.22), 1.69 (AOR = 1.69, 95% CI: 1.56, 1.85), 6.75 (AOR = 6.75, 95% CI: 6.14, 7.43), 1.06 (AOR = 1.06, 95% CI: 0.97, 1.15) times more odds of participating in decision making for their health care utilization respectively as compared to women in Chad. On the other hand the odds of participating in decision making among women in Burkina Faso, Mali and Niger were 19% (AOR = 0.81, 95% CI: 0.74, 0.88), 41% (AOR = 0.59, 95% CI: 0.54, 0.65), and 11% (AOR = 0.89, 95% CI: 0.82, 0.98) times less likely as compared to women in Chad, respectively (Table 3).

## Measures of variation

According to the result in the null model, the ICC was 38.9% of the variations in health care decision-making autonomy among study subjects that were attributed to differences at the cluster level, but the rest, 61.1%, were attributed to individual factors.

The final model indicates that the proportional change in variance (PCV) value of 0.472 indicates that both individual and community-level factors were responsible for about 47.2% of the variation in health care decision-making autonomy between study subjects. When comparing models and fitness, deviance was used, and this model was the best-fitting one, having

**Table 3. Multilevel analysis of factors associated with women's decision making autonomy on health care utilization among married women in high fertility sub-Saharan Africa countries.**

| Variables | Categories | Null model | Model 2 AOR (95% CI) | Model 3 AOR (95% CI) | Model 4 AOR (95% CI) |
|---|---|---|---|---|---|
| Age in years | 15–24 | | 1 | | 1 |
| | 25–34 | | 1.37 (1.34, 1.43) | | 1.34(1.29,1.39)** |
| | 35 and above | | 1.86 (1.78, 1.93) | | 1.82(1.75,1.90)** |
| Women education status | No education | | 1 | | 1 |
| | Primary | | 1.65 (1.58, 1.71) | | 1.25(1.20, 1.31)** |
| | Secondary and Higher | | 1.75 (1.67, 1.83) | | 1.44(1.37,1.52)*** |
| Husband education status | No education | | 1 | | 1.00 |
| | Primary | | 1.81 (1.73, 1.88) | | 1.24(1.18,1.29)* |
| | Secondary and Higher | | 1.53 (1.47, 1.59) | | 1.21(1.15,1.27)** |
| Occupation of women | Not working | | 1 | | 1 |
| | Working | | 1.69 (1.63, 1.75) | | 1.67(1.59,1.74)* |
| Household size | 1–2 | | 1 | | 1.00 |
| | 3–5 | | 1.05 (0.98, 1.12) | | 0.96(0.89,1.03) |
| | 6 and above | | 0.97 (0.91, 1.04) | | 0.87(0.81,1.00) |
| Sex of household head | Male | | 1 | | 1 |
| | Female | | 1.70 (91.63, 1.78) | | 1.44 (1.37,1.51)** |
| Media exposure | No | | 1 | | 1 |
| | Yes | | 1.12 (1.08, 1.15) | | 1.04(1.09,1.18)** |
| Covered by health insurance | Yes | | 1.12 (1.08, 1.15) | | 1.26(1.17,1.36)** |
| | No | | 1 | | 1 |
| Residence | Rural | | | 1 | 1 |
| | Urban | | | 1.34 (1.29,1.39) | 1.14(1.09,1.19)* |
| Community media exposure | Low | | | 1 | 1 |
| | High | | | 2.26(1.92, 2.68) | 1.93(1.65,2.25)* |
| Community level education | Low | | | 1 | 1 |
| | High | | | 3.05 (2.57, 3.63) | 2.43(2.07, 2.85)** |
| Community level poverty | Low | | | 1.33 (1.11,1.58) | 1.27(1.08, 1.49)* |
| | High | | | 1 | 1 |
| Countries | | | | | |
| | Angola | | | 9.00 (8.39, 9.66) | 6.44 (5.87, 7.06)** |
| | Burkina Faso | | | 0.94 (0.88, 1.02) | 0.81 (0.74, 0.88)* |
| | Burundi | | | 8.86 (8.33, 9.44) | 5.69 (5.21, 6.22)* |
| | DR Congo | | | 2.64 (2.49, 2.80) | 1.69 (1.56, 1.85)** |
| | Gambia | | | 7.16 (6.65, 7.69) | 6.75 (6.14, 7.43)** |
| | Mali | | | 0.68 (0.63, 0.74) | 0.59 (0.54, 0.65)* |
| | Nigeria | | | 1.53 (1.45, 1.64) | 1.06 (0.97, 1.15)* |
| | Niger | | | 0.81 (0.75, 0.86) | 0.89 (0.82, 0.98)* |
| | Chad | | | 1 | 1 |
| Random effect | | | | | |
| Variance | | 2.10 | 1.22 | 1.34 | 1.10 |
| ICC (%) | | 38.9 | 27.11 | 28.5 | 25.22 |
| MOR | | 3.96 | 2.85 | 3.00 | 2.71 |
| PCV | | Ref | 41.9 | 36.19 | 47.61 |
| Model comparison | | | | | |
| Deviance | | 141123.1 | 122710 | 123289.3 | 111164.8 |

*(Continued)*

**Table 3.** (Continued)

| Variables | Categories | Null model | Model 2 AOR (95% CI) | Model 3 AOR (95% CI) | Model 4 AOR (95% CI) |
|---|---|---|---|---|---|
| VIF | | Ref | 2.06 | 2.09 | 1.53 |

* = P-value < 0.05,

** = P_value < 0.01,

*** = P_value < 0.001,

ICC = Intra class correlation coefficient, MOR = Median odds ratio, PCV = proportional change in variance. AOR = adjusted odds ratio; CI = confidence interval,

VIF = Variance inflation factor

the lowest deviance (111164.8). The variance inflation factor (VIF) was calculated to check the effect of multi-co linearity and yielded a mean value of 1.53 for the final model (Table 3).

## Discussion

Women's decision-making autonomy on their health care utilization is an essential component of sexual and reproductive health rights [34]. This study aimed to determine women's decision making autonomy on health care utilization and associated factors among reproductive age women in high fertility countries in SSA. As a result, 42.03% of women in high fertility Sub-Saharan Africa countries have autonomy in deciding how to use healthcare, with Mali having the lowest prevalence of 19.74%. This implies that the majority (six in ten) women had no autonomy to decide for their own health care utilization. This in turn affects lower total fertility reduction and higher infant mortality [10].

The associated factors of women's decision-making autonomy on health care utilization were identified as women's age, educational status of the women, husband's education, sex of household head, media exposure, health expenditure covered by health insurance, residence, community level media exposure, community level poverty, community level education, and country. This finding is higher than the study conducted in Nigeria 38.9% [35] but lower than the study reported in Ethiopia 81.6% [21] and Ghana 75% [36]. This might be due to the large sample size (comprehensive nature of our study) and the fact that we included participants from different countries with different socioeconomic backgrounds. Additionally, the difference may account that different countries may take different measures and strategies to empower women.

Women's participation in healthcare utilization decision-making increased with age; respondents aged 25–34 and 35 or older were more likely to have decision-making autonomy than younger age groups. It is consistent with the studies conducted in Ethiopia [37, 38], and Indonesia [39], which indicated that the percentage of women who usually participate in decision making increases with age. This might be because women's positions in society are socially constructed, and their status varies depending on their age and role in society [40]. Additionally, in many African societies, as a woman gets older, she becomes more autonomous since self-esteem increases with age [39]. The findings suggest that as women age increased, they will make more autonomous decisions regarding their health care.

Women who completed formal education had higher odds of health care decision-making autonomy compared to women with no formal education. This is consistent with study findings in Ethiopia [37], sub-Saharan Africa [41], and Asia [42]. This might be because education enhances self-confidence, their knowledge towards health care decision-making, and build up women's capacity in developing their own decisions. In addition, educated women are more

likely to take part in decision making in their health care because they exercised gender equality [43, 44].

Women whose partners had completed formal education were more likely to participate in health care decision-making compared to women whose partners had no formal education. This is in agreement with studies conducted in Ethiopia [37, 38], Nigeria [35], and SSA [41]. The possible explanation might be that if partner is educated, the more he will accept gender equality and believe in equal participation in decision making [38, 41]. Hence, education plays an important role in influencing later-life decision-making about maternal healthcare utilization.

This study also found that women's autonomy in health-care decision-making was associated with higher odds among urban residents than rural residents. This is in line with studies conducted in Ethiopia [45], Bangladesh [46], and Nigeria [35]. The possible justification might be that urban women are more educated and have better exposure to mass media. Because among women those who are educated and exposed to mass media are more likely active in decision making [42, 47]. Moreover urban women are usually challenges the traditional belief of male dominance in household decision making [48].

This study's findings show that women occupation was found to be a significant factor in the health care decision-making autonomy of women. This study found that women who are working had a higher chance of participating in health care decision-making than women who are not working. This is in line with studies in Ethiopia [21], Ghana [36], Nepal [45], and Asia [42] which found that working women are more likely to participate in health care decision-making than women who are not working. This could be because women who are working have more education and contact with different individuals; this might enhance women's decision-making confidence and access to information, and challenge the beliefs of men's dominance in making decisions [21, 43]. On the contrary studies conducted in Ethiopia [49] and South Africa [50] revealed that women who had no work were more autonomous in health care decision as compared with those women who had work. It would therefore be appropriate to conduct further qualitative assessments in order to investigate such odd seeming results.

The association between the sex of the household head and women's autonomy in making decisions about health care utilization was positively associated, in which women who were living in a household with a female household head were more likely to participate in health care decision making than their counterparts. One possible justification is that being the head of the household allows women to make decisions and take responsibility on their own, and their decisions may be less influenced by others [51]. This implies that sex of the household can provide insights into gender inequality and power dynamics that exist at the household level [52].

Women with a high media exposure were more likely to have decision-making autonomy on their health care utilization compared to their counterparts. This finding is consistent with the results of studies done in Ethiopia [53], Nigeria [35], and Pakistan [54]. The possible reason for this finding is women with high media exposure might have a better understanding of reproductive health rights and the advantages of their health care service utilization that encourages their participation in reproductive health decisions [53]. Therefore media exposure is considered as one of the best strategy for women's autonomy in health care utilization.

The odds of decision-making autonomy on health care utilization among women whose healthcare services were covered by health insurance were higher as compared to their counterparts. This might be because out-of-pocket payment for services is one of the constraints to deciding and utilizing health care services [55]. Furthermore, their decision is likely to be influenced by the relative total worth of expected expenses in obtaining necessary health care. This study also found that women who lived in high-income communities had more autonomy in their decision-making when it came to health care utilization than women who lived in low-

income communities. This finding is supported by studies conducted in Ethiopia [38] and Nepal [44]. This might be because women living in low-community level poverty are participating in income-generating activities, which increases women's economic independence and challenges the traditional belief of male dominance in decision-making, hence improving spousal communication in household decision-making autonomy [56].

The odds of women's autonomy on health care utilization was higher among women who lived in Angola, Burundi, DR Congo, Gambia, and Nigeria than in Chad. On the other hand, the autonomy of women who lived in Burkina Faso, Niger and Mali were less odds of health care decision making than in Chad. This might be because the contributions and commitments of each country's government to empowering women may vary from one to another [57].

## Strengths and limitations

The study's main strength was that it used nationally representative survey data with a large sample size. As the data are hierarchical, we used a multilevel analysis (a more advanced model). However, the data used in this study are cross-sectional, which limits the conclusions about the causality of factors in the dependent variable. Moreover, DHS surveys are based on self-reported information, which is likely to be prone to social desirability bias due to its socio-cultural nature.

## Conclusion

Despite the fact that every woman has the right to make her own health care decisions, 58% of them had no role in decisions about their health care utilization. Age of the women, educational status of women, husband's education, occupation of the respondent, sex of household head, media exposure, health expenditure covered by health insurance, residence, and community education were the factors associated with women's decision-making autonomy on their health care utilization. The government should promote women's decision-making autonomy on their health care utilization as essential components of sexual and reproductive health rights through education and mass media, with particular attention for adolescent women, women living in rural Moreover, promoting sexual and reproductive health education targeting school-age women is essential to the realization of sexual and reproductive health rights.

## Acknowledgments

We are grateful to the DHS programs, for the permission to use all the relevant DHS data for this study.

## Author Contributions

**Conceptualization:** Wubshet Debebe Negash, Desale Bihonegn Asmamaw.

**Data curation:** Wubshet Debebe Negash, Getachew Teshale Kefale, Tadele Biresaw Belachew, Desale Bihonegn Asmamaw.

**Formal analysis:** Wubshet Debebe Negash, Desale Bihonegn Asmamaw.

**Methodology:** Wubshet Debebe Negash.

**Software:** Wubshet Debebe Negash, Tadele Biresaw Belachew.

**Supervision:** Getachew Teshale Kefale.

**Validation:** Wubshet Debebe Negash, Getachew Teshale Kefale.

**Visualization:** Wubshet Debebe Negash, Getachew Teshale Kefale.

**Writing – original draft:** Wubshet Debebe Negash, Getachew Teshale Kefale, Desale Biho-negn Asmamaw.

**Writing – review & editing:** Wubshet Debebe Negash, Getachew Teshale Kefale, Tadele Bire-saw Belachew.

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
