## [Decision Letter · Decision Letter 0]

29 May 2023

PONE-D-23-11388Married women decision-making autonomy on health care utilization in high fertility sub-Saharan African countries: a multilevel analysis of recent Demographic and Health SurveyPLOS ONE

Dear Dr. Negash,

Thank you for submitting your manuscript to PLOS ONE. After careful consideration, we feel that it has merit but does not fully meet PLOS ONE’s publication criteria as it currently stands. Therefore, we invite you to submit a revised version of the manuscript that addresses the points raised during the review process.

We look forward to receiving your revised manuscript.

Kind regards,

Obasanjo Afolabi Bolarinwa, Masters

Academic Editor

PLOS ONE

Journal Requirements:

https://bmchealthservres.biomedcentral.com/articles/10.1186/s12913-017-2670-9

https://www.unipid.fi/virtual-studies/globalisation-and-corporate-responsibility/

https://ssphfrontiersfiles.blob.core.windows.net/articles/files/1604905/pubmed-zip/.versions/4/.package-entries/ijph-67-1604905-r3/ijph-67-1604905.pdf?

https://bmjopen.bmj.com/content/12/5/e059307.full

https://so03.tci-thaijo.org/index.php/jpss/article/view/168910

https://www.researchgate.net/publication/341236214_Women's_Decision_on_Contraceptive_Use_in_Ethiopia_Multinomial_Demographic_and_Health_Surve

In your revision ensure you cite all your sources (including your own works), and quote or rephrase any duplicated text outside the methods section. Further consideration is dependent on these concerns being addressed.

   "This research has received no specific grant from any funding agency in the public, commercial or not for profit sectors"

Additional Editor Comments:

This manuscript has been reviewed and there are minor corrections to be made. Kindly revise the manuscript in line with the reviewers’ comments.

Reviewers' comments:

Reviewer's Responses to Questions

**Comments to the Author**

1. Is the manuscript technically sound, and do the data support the conclusions?

Reviewer #1: Yes

Reviewer #2: Yes

2. Has the statistical analysis been performed appropriately and rigorously? 

Reviewer #1: No

Reviewer #2: Yes

3. Have the authors made all data underlying the findings in their manuscript fully available?

Reviewer #1: Yes

Reviewer #2: No

4. Is the manuscript presented in an intelligible fashion and written in standard English?

Reviewer #1: No

Reviewer #2: No

5. Review Comments to the Author

Reviewer #1: General Comment: Although the paper is well written paper, there are a lot of grammatical errors that need to be attended to. I have pointed out a few to the authors, there are many to be attended to. Generally revision should be needed.The authors have done a good work. Also, the authors need to be consistent with their decimal separator.

Abstract

Authors need to indicate the lowest and highest prevalent countries

Introduction

Line 58: it should be depend not depends

Line 60: Authors need to support this statement

Line 61: Authors indicate findings implied that. Can authors state the region of these findings

Line 69: Is education status not a component of socioeconomic position. Authors can say "Women’s decision with regard to the use of healthcare services may depend on their economic position, perception towards modern healthcare treatment, educational status, family size, perceived severity of illness, and previous experience of illness"

Line 72: the aforementioned factors "have" frequently demarcated

Line 73: Authors indicate "Additionally, various findings......but state only a source

Line 78: Authors can add "most" to .....in the countries of SSA

Line 84: Statement not clear

Methods

Study settings

Line 98: delete "was"

Line 101: replace Africa with SSA unless authors want to refocus their study area to Africa

Line 130: Authors should introduce "On" not on

Line 130-131: It's confusing

Lines 224-226: should be relooked at

The authors did not indicate if all the variables were statistically significant for the regression analysis. The p-values for the variables were not stated. This can be added to Table 2

Authors should mention the mathematical formula how to compute ICC and others?

Discussion

What are the implications of the prevalence in this study?

The discussion section needs to be more focused on the implications

Authors need to get literature that do not support their findings as well

Line 284: What is the justification for the comprehensive nature of this study?

Line 294: were should be deleted

Recommendation

Please minimize your recommendation hence the current one is too long

Reviewer #2: First, I would like to appreciation for studying this interesting topic. Overall, the manuscript is good but I would like to forward my concerns and suggestions as follows,

The authors should improve the quality of the writing throughout their manuscript. The manuscript would benefit from proof-reading for grammatical and typographical errors.

The authors should provide some details on the justification of the study in terms of the study area (why you include high fertility sub-Saharan African countries and the implication of the problem)

In the discussion section line 282-285, the author compared the findings with previous studies in terms of similarity or difference. However, the underlying causes for the discrepancies between studies need to be elaborated to increase the depth of the discussion and to allow readers to know more about the situation. Also, the authors should mention the public health importance of in the finding.

The study claimed to take a multi-level approach in its statistical analysis. However, the idea of a ‘multi-level’ model in the paper seems somewhat different from classical multi-level models in the statistics literature. The fixed and random effects models required a more detailed concept about the model for better clarity.

Line 182-184 , the authors write about terms like ICC, MOR in which the readers may not be familiar to readers who did not have a technical background. It is important to provide a simple and intuitive definition of such technical terms before they are applied.

6. PLOS authors have the option to publish the peer review history of their article (what does this mean?). If published, this will include your full peer review and any attached files.

Reviewer #1: No

Reviewer #2: No

---

## [Author Response · Author response to Decision Letter 0]

5 Jun 2023

Date: June, 2023

To: PLOSE ONE Journal Editorial Office

Subject: Submitting a revised manuscript for publication

Dear Editor,

First of all, we would like to thank you for the opportunity given to revise our manuscript for improvement. We are writing to submit the revised version for consideration of publication in PLOSE ONE journal. We have addressed all reviewers and editorial queries raised. Grammar and spellings have been improved throughout the manuscript by English language experts, and we have made rewording and rephrasing some parts of the paragraphs of the paper accordingly. We also ensured you that the track change option of MS Word has been used to indicate modifications made on the manuscript. Finally, we are submitting a revised article entitled “Married women decision making autonomy on health care utilization in high fertility sub-Saharan African countries: a multilevel analysis of recent Demographic and Health Survey” to your editorial office for consideration of it for publication. We are very grateful to both the editor and reviewers for your comments and suggestions. All the concerns raised so far will have an undeniable impact on improving the quality and readability of our scholarly work. Appreciating your effort and valuable comments, we have provided possible reflections and amended the raised concerns and questions. Kindly find our reflections here.

We hope you will consider the revised manuscript acceptable for publication in PLOSE ONE research journal. We have also submitted point by point responses for editor’s and reviewers’ comments.

Authors’ Point-by-point response to editor and reviewer comments 

We are very grateful to both the editor and reviewers for your comments and suggestions. All the concerns raised so far will have an undeniable impact on improving the quality and readability of our scholarly work. Appreciating your effort and valuable comments, we have provided possible reflections and amended the raised concerns and questions. Kindly find our reflections here.

We hope you will consider the revised manuscript acceptable for publication in PLOSE ONE research journal.

S.no Editor comments Authors’ responses

1 This manuscript has been reviewed and there are minor corrections to be made. Kindly revise the manuscript in line with the reviewers’ comments. Dear editor we are grateful for our comments and prompt responses. We appreciate all your contributions for the scientific world. We accepted all the comments and corrected all the editor and reviewer comments. 

 Reviewer comments 

 Reviewer #1 

1 General Comment: Although the paper is well written paper, there are a lot of grammatical errors that need to be attended to. I have pointed out a few to the authors, there are many to be attended to. Generally revision should be needed. The authors have done a good work. Also, the authors need to be consistent with their decimal separator. Dear reviewer, your observations and insightful comments are too important. With regard to the grammatical errors, yes you are correct there may be grammatical errors. We revised the whole body of the manuscript to minimize the grammatical errors. Again we revised the decimal separator. Kindly see the clean version of the manuscript. 

2 Abstract

Authors need to indicate the lowest and highest prevalent countries Thank you for the input. We added the highest and the lowest prevalent countries in the abstract. Kindly see the clean version of the manuscript at page 3 line 34-35

 Introduction 

3 Line 58: it should be depend not depends Thank you. We revised and write it as “depend”. Kindly see the clean version of the manuscript page 5 line 59,

4 Line 60: Authors need to support this statement Thank you for the observation. Supported by reference. Kindly see the clean version of the manuscript at page 5 line 61. 

5 Line 61: Authors indicate findings implied that. Can authors state the region of these findings Yes we can.

We stated the region at page 5 line 62.

6 Line 69: Is education status not a component of socioeconomic position. Authors can say "Women’s decision with regard to the use of healthcare services may depend on their economic position, perception towards modern healthcare treatment, educational status, family size, perceived severity of illness, and previous experience of illness" Of course education is one of the socioeconomic component. We revised as “Women’s decision with regard to the use of healthcare services may depend on their economic position, perception towards modern healthcare treatment, family size, perceived severity of illness, and previous experience of illness” kindly see the clan version of the manuscript at page 5-6 line 71-76.

7 Line 72: the aforementioned factors "have" frequently demarcated Thank you in advance for the important input. We add the word “have”. Kindly refer the clean version of the manuscript at page 6 line 73.

8 Line 73: Authors indicate "Additionally, various findings......but state only a source Thank you very much. We revised it. Kindly refer the revised manuscript on Page 6 Line 75-76. 

9 Line 78: Authors can add "most" to.....in the countries of SSA The word “most” is added based on your important suggestions. Kindly see the revised manuscript on page 6 line 80.

10 Line 84: Statement not clear Thank you for your question we made it clear. Kindly refer the clean version of the manuscript page 6 line 86-87.

 Methods Study settings 

11 Line 98: delete "was" Deleted. Page and line: N/A

12 Line 101: replace Africa with SSA unless authors want to refocus their study area to Africa Replaced. Kindly find in the clean version of the manuscript on page 7 line 107.

13 Line 130: Authors should introduce "On" not on Thank you very much. We had revised based on the comment. Kindly see the revised manuscript on page 9 line 135-137.

14 Line 130-131: It's confusing Revised and made clear. Kindly see the clean version of the manuscript page 9 line 135-137.

15 Lines 224-226: should be relooked at Thank you very much. We looked it and take correction too. Kindly see the clean version of the manuscript page 16 line 216-219. 

16 The authors did not indicate if all the variables were statistically significant for the regression analysis. The p-values for the variables were not stated. This can be added to Table 2 Dear reviewer we indicated the statistically significant variables in table 3. Kindly refer table 3.

17 Authors should mention the mathematical formula how to compute ICC and others? Dear reviewer thank you for your suggestion. Now we stated all the formulas accordingly. Kindly refer page 12 line 204-216.

 Discussion 

18 What are the implications of the prevalence in this study? The implication of the prevalence of this study is indicated. Kindly see page 21 line 310-311.

19 The discussion section needs to be more focused on the implications Of course we accepted that the discussion should to include more about the implications of the study. The manuscript is revised as per the given comments. Kindly refer the manuscript page 21 line 310-312 page 22 line 329-330, 342-343; page 23 line 361-362, 368-370; page 24 line 376-377.

20 Authors need to get literature that do not support their findings as well Yes, it is recommended to discuss the findings with other researches that do not support our finding as well. We tried to find this and included it. For example you can refer the clean version of the manuscript at Page 23 line 359-362.

21 Line 284: What is the justification for the comprehensive nature of this study? Dear reviewer, the reason why we say comprehensive is because we took the nationally representative data of each high fertility country and pooled together to understand the magnitude of decision making autonomy in health care. Page and line= N/A

22 Line 294: where should be deleted Thank you in advance once again for your key comments. We deleted were. Page and line=N /A

 Recommendation

Please minimize your recommendation hence the current one is too long Thank you for your key observation we accepted your comments and made the recommendation minimized. Kindly see the clean version of the manuscript page 4 line 47-49 and page 25 line 407-411

 Reviewer 2 

1 First, I would like to appreciation for studying this interesting topic. Overall, the manuscript is good but I would like to forward my concerns and suggestions as follows, Dear reviewer thank you for your appreciation. We positively accept all your comments and respond all the questions accordingly.

2 The authors should improve the quality of the writing throughout their manuscript. The manuscript would benefit from proof-reading for grammatical and typographical errors. Again thank you for your academic suggestion. We accepted your direction and proofread the manuscript.

3 The authors should provide some details on the justification of the study in terms of the study area (why you include high fertility sub-Saharan African countries and the implication of the problem) Dear reviewer thank you for the key observation. We have tried to add justification/reason why we focus on the setting of high fertility country. Kindly refer page 6-7 line 86-98.

4 In the discussion section line 282-285, the author compared the findings with previous studies in terms of similarity or difference. However, the underlying causes for the discrepancies between studies need to be elaborated to increase the depth of the discussion and to allow readers to know more about the situation. Also, the authors should mention the public health importance of in the finding. We accepted your comments. Based on the comment the authors are tried to address the underline causes of the discrepancies. Kindly refer page 21 line 318-321.

5 The study claimed to take a multi-level approach in its statistical analysis. However, the idea of a ‘multi-level’ model in the paper seems somewhat different from classical multi-level models in the statistics literature. The fixed and random effects models required a more detailed concept about the model for better clarity. Of course the authors putted detail explanation for model clarity. Kindly refer the clean version of the manuscript on page 11-12 line 182-200.

6 Line 182-184 the authors write about terms like ICC, MOR in which the readers may not be familiar to readers who did not have a technical background. It is important to provide a simple and intuitive definition of such technical terms before they are applied. That is very important comment. We accepted and provided detailed explanation. Kindly refer the clean version of the manuscript page 12 line 204-216.

Kind regards,

Wubshet Debebe Negash

Corresponding author

---

## [Decision Letter · Decision Letter 1]

2 Jul 2023

Married women decision making autonomy on health care utilization in high fertility sub-Saharan African countries: a multilevel analysis of recent Demographic and Health Survey

PONE-D-23-11388R1

Dear Dr. Wubshet,

We’re pleased to inform you that your manuscript has been judged scientifically suitable for publication and will be formally accepted for publication once it meets all outstanding technical requirements.

Kind regards,

Obasanjo Afolabi Bolarinwa, Masters

Academic Editor

PLOS ONE

Additional Editor Comments (optional):

Reviewers' comments:

Reviewer's Responses to Questions

**Comments to the Author**

1. If the authors have adequately addressed your comments raised in a previous round of review and you feel that this manuscript is now acceptable for publication, you may indicate that here to bypass the “Comments to the Author” section, enter your conflict of interest statement in the “Confidential to Editor” section, and submit your "Accept" recommendation.

Reviewer #1: All comments have been addressed

Reviewer #2: All comments have been addressed

2. Is the manuscript technically sound, and do the data support the conclusions?

Reviewer #1: Yes

Reviewer #2: Yes

3. Has the statistical analysis been performed appropriately and rigorously? 

Reviewer #1: Yes

Reviewer #2: Yes

4. Have the authors made all data underlying the findings in their manuscript fully available?

Reviewer #1: Yes

Reviewer #2: Yes

5. Is the manuscript presented in an intelligible fashion and written in standard English?

Reviewer #1: No

Reviewer #2: (No Response)

6. Review Comments to the Author

Reviewer #1: Authors have done a good job. However, I still think that it will need another round of English editing as there are some few mistakes.

Reviewer #2: “Married women decision making autonomy on health care utilization in high fertility sub-Saharan African countries: a multilevel analysis of recent Demographic and Health Survey”

Review decision: Accept

The paper is based on a good broad study of married women decision making autonomy on health care utilization in high fertility sub-Saharan African countries: a multilevel analysis of recent Demographic and Health Survey

Review decision: Accept

The authors address my comments.

7. PLOS authors have the option to publish the peer review history of their article (what does this mean?). If published, this will include your full peer review and any attached files.

Reviewer #1: No

Reviewer #2: **Yes: **Fantu Mamo Aragaw

---

## [Editor Report · Acceptance letter]

6 Jul 2023

PONE-D-23-11388R1 

Married women decision making autonomy on health care utilization in high fertility sub-Saharan African countries: a multilevel analysis of recent Demographic and Health Survey 

Dear Dr. Negash:

I'm pleased to inform you that your manuscript has been deemed suitable for publication in PLOS ONE. Congratulations! Your manuscript is now with our production department. 

Kind regards, 

on behalf of

Dr. Obasanjo Afolabi Bolarinwa 

Academic Editor

PLOS ONE